# Development of the NOGGO GIS v1 Assay, a Comprehensive Hybrid-Capture-Based NGS Assay for Therapeutic Stratification of Homologous Repair Deficiency Driven Tumors and Clinical Validation

**DOI:** 10.3390/cancers15133445

**Published:** 2023-06-30

**Authors:** Eva-Maria Willing, Claudia Vollbrecht, Christine Vössing, Peggy Weist, Simon Schallenberg, Johanna M. Herbst, Stefanie Schatz, Balázs Jóri, Guillaume Bataillon, Philipp Harter, Vanda Salutari, Antonio Gonzáles Martin, Ignace Vergote, Nicoletta Colombo, Julia Roeper, Tobias Berg, Regina Berger, Bettina Kah, Trine Jakobi Noettrup, Markus Falk, Kathrin Arndt, Andreas Polten, Isabelle Ray-Coquard, Franziska Selzam, Judith Pirngruber, Stefanie Schmidt, Michael Hummel, Markus Tiemann, David Horst, Jalid Sehouli, Eric Pujade-Lauraine, Katharina Tiemann, Elena Ioana Braicu, Lukas C. Heukamp

**Affiliations:** 1Institut für Hämatopathologie Hamburg, 22547 Hamburg, Germanymtiemann@hp-hamburg.de (M.T.); heukamp@hp-hamburg.de (L.C.H.); 2Nord-Ostdeutsche Gesellschaft für Gynäkologische Onkologie NOGGO e. V., 13359 Berlin, Germany; 3Institute of Pathology, Charité-Universitätsmedizin Berlin, Corporate Member of Freie Universität Berlin and Humboldt-Universität zu Berlin, 10117 Berlin, Germany; 4Lungenkrebsmedizin Oldenburg, GbR, 26121 Oldenburg, Germany; 5Department of Pathology, Institut Universitaire du Cancer de Toulouse (IUCT) Oncopole, 31100 Toulouse, France; 6Kliniken Essen Mitte, 45276 Essen, Germany; 7Department of Woman, Child and Public Health, Fondazione Policlinico Universitario A. Gemelli IRCCS, Largo Agostino Gemelli 8, 00168 Rome, Italy; 8Medical Oncology Department, Clinica Universidad de Navarra, 28027 Madrid, Spain; 9GEICO, 28003 Madrid, Spain; 10Leuven Cancer Institute, University Hospital Leuven, 3000 Leuven, Belgium; 11Gynecologic Oncology Program, European Institute of Oncology, 20141 Milan, Italy; 12Department of Medicine and Surgery, University of Milan-Biocca (Colombo), 20141 Milan, Italy; 13Universitätsklinik für Innere Medizin-Onkologie, Cancer Center Oldenburg, Pius-Hospital, Georgstr. 12, 26121 Oldenburg, Germany; 14Department of Obstetrics and Gynecology, AGO Austria Study Center, Medical University Innsbruck, 6020 Innsbruck, Austria; 15Copenhagen University Hospital, 2100 Copenhagen, Denmark; 16Agilent Technologies Deutschland GmbH, 71034 Böblingen, Germany; andreas_polten@agilent.com; 17Centre Léon BERARD, and University Claude Bernard Lyon I, 69008 Lyon, France; 18ARCAGY GINECO, 75008 Paris, France; 19German Cancer Consortium (DKTK) Partner Site Berlin, German Cancer Research Center (DKFZ), 69120 Heidelberg, Germany; 20Charité University Medicine, Joint Medical Faculty of Freie Universität Berlin and Humboldt-Universität zu Berlin, Department of Gynecology with Center of Oncological Surgery and European Competence Center for Ovarian Cancer, 10117 Berlin, Germany; 21Department of Obstetrics and Gynecology, Stanford University, Stanford, CA 94305, USA

**Keywords:** HRD, homologous repair deficiency, HRR, genomic instability, *BRCA1*, *BRCA2*, ovarian cancer, PAOLA-1, PFS, OS, breast cancer, somatic mutation, LOH, molecular pathology, diagnostics, DNA repair, PARP inhibition, PARPi

## Abstract

**Simple Summary:**

Genomic instability (GI) caused by homologous repair deficiency (HRD) is a novel highly clinically relevant biomarker that cannot only identify patients suffering from high-grade serous ovarian cancer that may benefit from poly-ADP ribose polymerase (PARP) inhibitors but also helps in functionally annotating mutations found within genes of the homologous repair pathway. Tumors in which GI plays a role in therapeutic considerations currently include ovarian, breast, prostate, and pancreatic cancer. Therefore, we developed, implemented, and clinically validated a comprehensive custom Agilent XT HS2 hybrid capture next-generation sequencing (NGS) assay that allows in addition to the analysis of homologous recombination repair (HRR) pathway and relevant cancer genes, complex *BRCA1* and *BRCA2* alterations including large deletions and the evaluation of the GI-Score (GIS) status on one single tumor sample. The NOGGO (Northeastern German Society for Gynecologic Oncology) GIS v1 assay was validated as part of the European Network for Gynaecological Oncological Trial groups (ENGOT) HRD European Initiative on a subset of the ENGOT PAOLA-1 clinical trial samples. Patients identified as HRD-positive using the NOGGO GIS v1 assay showed a benefit of progression-free survival (PFS) and overall survival (OS) with comparable hazard ratios to the Myriad MyChoice assay.

**Abstract:**

The worldwide approval of the combination maintenance therapy of olaparib and bevacizumab in advanced high-grade serous ovarian cancer requires complex molecular diagnostic assays that are sufficiently robust for the routine detection of driver mutations in homologous recombination repair (HRR) genes and genomic instability (GI), employing formalin-fixed (FFPE) paraffin-embedded tumor samples without matched normal tissue. We therefore established a DNA-based hybrid capture NGS assay and an associated bioinformatic pipeline that fulfils our institution’s specific needs. The assay´s target regions cover the full exonic territory of relevant cancer-related genes and HRR genes and more than 20,000 evenly distributed single nucleotide polymorphism (SNP) loci to allow for the detection of genome-wide allele specific copy number alterations (CNA). To determine GI status, we implemented an %CNA score that is robust across a broad range of tumor cell content (25–85%) often found in routine FFPE samples. The assay was established using high-grade serous ovarian cancer samples for which *BRCA1* and *BRCA2* mutation status as well as Myriad MyChoice homologous repair deficiency (HRD) status was known. The NOGGO (Northeastern German Society for Gynecologic Oncology) GIS (GI-Score) v1 assay was clinically validated on more than 400 samples of the ENGOT PAOLA-1 clinical trial as part of the European Network for Gynaecological Oncological Trial groups (ENGOT) HRD European Initiative. The “NOGGO GIS v1 assay” performed using highly robust hazard ratios for progression-free survival (PFS) and overall survival (OS), as well a significantly lower dropout rate than the Myriad MyChoice clinical trial assay supporting the clinical utility of the assay. We also provide proof of a modular and scalable routine diagnostic method, that can be flexibly adapted and adjusted to meet future clinical needs, emerging biomarkers, and further tumor entities.

## 1. Introduction

Defects of DNA repair mechanisms are one of the hallmarks of cancer and can offer therapeutic approaches for cancer patients. DNA defects are recognized by a spectrum of proteins leading to restoration of the DNA integrity or elimination of the damaged cells [1,2]. Prevalent single-strand breaks induced by oxygen radicals, chemotherapy, ionizing radiation, or oxygen radicals can be repaired by base excision repair (BER) mechanisms, including PARP proteins as major players [1,3]. Inhibition of PARP may induce the accumulation of single-strand breaks resulting in highly lethal double-strand breaks (DSB), which would be normally repaired by the HRR pathway.

HRR uses a homologous DNA sequence as a template for error-free repair and involves a complex network of proteins including the key proteins *BRCA1* and *BRCA2* [2,4]. Hence, HRD leads to the accumulation of oncogenic mutations and massive GI [5,6]. Tumor cells with HRD are more sensitive to DNA-damaging agents that usually provoke HRR [2]. Furthermore, they can be targeted using the synthetic lethality principle by inhibiting the function of a gene (e.g., PARP) in the presence of an additional mutation (e.g., *BRCA1/2*)

The therapeutic relevance of HRD became apparent more than two decades ago when *BRCA1/2* germline mutations were shown to be responsible for most hereditary breast cancer cases [7,8]. Pre-clinical data revealed that *BRCA* deficient cells show up to 1000-fold higher response rates to PARP inhibitors (PARPi) than *BRCA* heterozygous or wild-type cells [9,10]. These data could broaden the spectrum of patients that benefits from additional testing [11,12,13,14,15].

The impact of these findings gained importance when it became apparent that this mechanism also functions in sporadic breast and ovarian tumors [16,17]. Even more, most of the *BRCA* wild-type tumors show a similar phenotype to the *BRCA*-mutated ones [17,18,19,20]. These include mutations in other members of the HRR pathway (e.g., *ARID1A, ATM, ATR, BARD1, CHEK1/2, FANCA, PALB2, RAD51*).

The functional interpretation of mutations in typical tumor suppressor genes is difficult as mutations need to be not only pathogenic, but inactivation of both alleles leads to a full functional loss. This can either happen via loss of heterozygosity (LOH), a second inactivating mutation in trans or in the case of *BRCA1* though inactivation by promoter methylation [21,22,23]. The functional effect of HRD can be seen as genomic scars such as a high degree of genome-wide LOH events, but also a high number of telomeric allelic imbalances (TAI) or large-scale state transitions (LST) [24].

Recently four phase III first-line studies, PAOLA-1/ENGOT-ov25, PRIMA/ENGOT-ov26/GOG-3012, VELIA/GOG-3005 and ATHENA–MONO/GOG-3020/ENGOT-ov45 have demonstrated that the addition of a PARPi to platinum-based therapy +/− bevacizumab improved PFS in advanced ovarian cancer patients [25,26,27,28]. The benefit was greater when the tumor was HRD-positive according to the Myriad MyChoice test, independently of *BRCA* status. The PAOLA-1 olaparib plus bevacizumab maintenance regimen was approved in USA/Europe/Japan for HRD-positive patients.

To identify *BRCA1*/*2* wild-type tumors that are HRD-positive and may benefit from DNA-damaging agents and PARPi, different approaches have been developed [18,22,29]. Currently, there are two commercial assays for HRD assessment that have been validated in prospective clinical trials [12,13]. MyChoice HRD (Myriad, Salt Lake City, UT, USA) is based on the number LOH, TAI and LST events assessed across the genome. A case is HRD-positive when the score is ≥42 or *BRCA*-mutated (PRIMA, PAOLA-1, VELIA, NOVA) [22,30]. In contrast, single-score Foundation Focus CDx *BRCA* LOH assay (Foundation medicine, Cambridge, MA, USA) detects genomic and somatic *BRCA* mutations and a percentage of the genome affected by LOH, by which samples are categorized as LOH-high if the score is ≥16 (ATHENA Mono, ARIEL2/3, QUADRA) and in reviewed Li et al. [13,19,26,31].

Beside these central lab-based tests, other approaches enabling the evaluation of large chromosomal rearrangements use a combination of high-throughput genomic profiling techniques like array-based comparative genomic hybridization, SNP genotyping, and NGS are under evaluation.

Here, we describe an NGS hybrid-capture biomarker assay developed by the NOGGO that detects mutations in *BRCA1*/*2* as well as further 55 HRR-relevant genes and structural alterations to establish a predictive GIS. The NOGGO GIS was tested and correlated to known MyChoice HRD scores and reflects the percentage of the genome affected by GI measured by deviation from chromosome-specific copy number.

## 2. Materials and Methods

Sample cohort: 131 pre-characterized clinical FFPE samples as well as a total of 469 samples of the ENGOT initiative were used [32,33,34].

Assay design: A panel based on hybrid capture enrichment XT HS2 chemistry (Agilent Technologies, Santa Clara, CA, USA) was designed targeting all exonic bases Covering all coding bases of all known transcripts accord to NCBI as well as a minimum of 10 bp flanking region of selected genes (Table 1) and the OneSeq CNV Backbone (Agilent Technologies), which enriches approximately 20.000 selected SNP equally distributed across all chromosomes. This corresponds to around six SNP/Mb. To achieve higher sequencing depths in genes for somatic mutation calling, three times as many capture probes were used to cover the coding regions of the genes compared with the regions containing SNP.

DNA quality was assessed using the tape station (Agilent Technologies), and 50 and 100 ng DNA derived from FFPE tissue regardless of quality was subjected to library preparation according to the manufacturer’s instructions (version C0, August 2020) for hybrid capture XT HS2 chemistry (Agilent Technologies). For PAOLA-1 trail samples no quality assessment was performed due to limited availability. Libraries were sequenced with 20 million read pairs using 80 bp read length (3 bp unique molecular identifier, 2 dark cycles, 75 bp template) on a Nextseq500 or NextSeq2000 instrument (Illumina, San Diego, CA, USA).

Mutation calling in the exonic regions of target genes was performed using a custom workflow generated in the CLC Workbench (Qiagen, Hilden, Germany). Briefly, reads were grouped by unique molecular identifiers (UMI), mapped to the hg19 reference, and consensus reads were built using UMI information. Local re-alignment was performed to refine alignment at regions with InDels and structural re-arrangements. This was followed by analysis with the “Low Frequency Variant Detection” module to call mutations. Mutations were annotated using ANNOVAR, amino acid change, and splice site effects, as well as dbSNP and ClinVar information (build 155) [35].

Functional annotation of the variants followed common guidelines [36,37]. Variant classification and interpretation relied on both publicly accessible and internal databases, which considered population frequencies and information from the published literature. The functional assessment of individual variants was also aided by CNA analysis conducted on tumor tissue. In this analysis, information on LOH was used to judge the pathogenicity of tumor suppressor genes including BRCA1/2 and HRR genes. For the latter, the identification of a mutation along with LOH in the same gene could be indicative of pathogenicity [36].

Detecting allele-specific copy number alterations.

All tools used to calculate the NOGGO GIS v1 score are publicly available and can be downloaded with the relevant manuals (see below)

Briefly, UMI was extracted using the tool ExtractUmisFromBam (Fgbio tools v1.5.0). Mapping was performed using bwa mem (v0.7.17-r1188) with default settings. PCR duplicates were identified using UmiAwareMarkDuplicatesWithMateCigar (Picard tools v2.23.1) and overlapping reads pairs were clipped using ClipBam (Fgbio tools v1.5.0). For the detection of allele specific can, we included the R package PureCN (v1.20.0) [38] in our pipeline in combination with Mutect (v1.1.4) for variant calling. The resulting pipeline does not require a matched normal sample for its calculations. Instead, a cohort of process-matched normal was generated according to the PureCN manual [38]. The cohort of normal samples is used to exclude regions with high variance in coverage from the analysis for each sample before the calculation of log2 ratios between the tumor and the best fitting process-matched normal is performed. PureCN requires allelic frequency information of germline SNP across the genome to detect allele specific copy numbers. Coverage log2 ratios and variant files were used as input for PureCN to determine purity, ploidy, and the allele specific copy number profile in the tumor sample.

NOGGO genomic instability score GIS was calculated with allele specific copy number profile and three measures of HRD based on PureCN output: percent loss of heterozygosity (PLOH), percent copy number alteration (PCNA), and percent telomeric copy number alterations (PTCNA). These allow for a variable amount of sequenceable territory due to varying DNA quality found in FFPE samples and achieve a low dropout rate. In order to obtain a single score, the sum of PLOH, PCNA, and PTCNA was calculated.

Percent loss of heterozygosity PLOH reflects the percentage of observed LOH across the sequenced genomic territory. LOH regions are only considered for PLOH when they are larger than 10 Mb and smaller than a complete chromosome arm.

Percent copy number alteration PCNA reflects the percentage of the genome that shows allele specific copy number alterations compared to the expected copy number. The expected copy number was defined as covering the largest part of the chromosome, and the total size of all regions larger than 10 Mb that deviate from this expected copy number were calculated. The determined sizes for each chromosome are added and divided by the total size of the observed genome, leading to the percentage of the genome deviating from the expected copy number. The observed genome is defined as the fraction of the genome that is sufficiently covered by enriched SNP and reads. Copy number alterations affecting a whole chromosome arm were not counted.

Percent telomeric copy number alterations PTCNA reflects the percentage of telomers showing copy number alteration compared to the expected copy number of the respective chromosome. Segments at the beginning of a chromosome are only considered to reflect the telomer of the chromosome if the start of the segment has a distance smaller than 5 Mb to the beginning of the chromosome. Segments at the end of a chromosome are only considered to reflect the telomer of the chromosome if the end of the segment has a distance smaller than 5 Mb to the expected end of the chromosome. Some telomers are not sufficiently assembled in the reference genome version hg19, and it cannot be determined if they are affected by copy number alterations (i.e., p-arm, chromosome 21). Only telomeric segments larger than 1 Mb are considered. CNA covering a whole chromosome arm is not counted.

## 3. Results

### 3.1. Assay Design and Performance

Assay performance was determined by coverage distribution, repeatability, and reproducibility using different input amounts (50 ng and 100 ng) of FFPE-derived DNA from normal tissue and clinical samples, respectively. Mean coverages, the fraction of unique reads, and off-target rates were reproducible between replicates and different inputs (Figure 1). As intended, the mean coverage in genes was at least twice as large compared with all targets (Figure 1). At least 250 unique reads covering one position were needed to call somatic mutation with a minimal allele frequency of 5% with 95% sensitivity at 95% statistical power. Figure 1D shows that at least 95% of exonic positions in the assay have a coverage of 250x or more unique reads, if reaching a mean coverage of more than 700 reads in genes. This can be observed for 50 ng and 100 ng DNA input. To minimize processing costs, we used a single cohort of reference normals rather than reprocessing the reference normals with each sample batch. This is only possible if the coverage across targets is reproducible. The observed correlation of coverage between targets of replicates was *p* > 0.98 (*p*-value < 1 × 10^−16^). However, for GC rich targets (%GC > 60%) GC bias can play a role (Figure 1F) even between replicates of the same run. However, the number of GC rich probes in the design is very limited (Figure 1E) and GC bias corrections were performed during copy number calculation, which guarantees stable copy number calling in case of slight GC bias.

### 3.2. Determining a Cutoff for the NOGGO GIS

In total, 131 DNA FFPE samples (including 27 duplicates) with known HRD status were available [33]. Of these, 96 (73%) were homologous repair proficient (HRP) (MyChoice score < 42) and 35 (27%) were homologous-repair-deficient (Myriad MyChoice score ≥ 42). Using this information as the true GI status, we determined a cutoff for the NOGGO GIS by maximizing Cohen’s kappa coefficient. In addition, we calculated sensitivity (95%), positive predictive value (PPV) (97%), specificity (98%), and negative predictive value (NPV) (99%) (Figure 2). The maximum kappa (k = 0.94) was reached using a cutoff between 83 and 85 for the NOGGO GIS. For further analyses, a cutoff of 83 was chosen to maximize the number of patients to benefit from PARP inhibition.

### 3.3. Clinical Validation of the NOGGO GIS v1 Assay

In phase 2 of the ENGOT HRD European initiative, the NOGGO GIS v1 assay underwent an evaluation. A total of 85 FFPE DNA samples from patients with wild-type *BRCA1/2* genes who participated in the PAOLA-1 trial were provided by the ENGOT and analyzed as described above in a blinded manner [32]. HRD status was determined with the NOGGO GIS v1 assay and was reported to the ENGOT for statistical analysis. When compared with the trial assay data, the results showed a Kappa value of 0.8 and a concordance rate of 90%. These results from the initial small cohort were deemed satisfactory to justify testing the assay in a much larger phase 3 cohort. The correlation to the Myriad MyChoice test on 85 tumor samples from PAOLA-1 *BRCA* wild-type patients using the KAPPA statistics is shown in Figure 3.

As part of phase 3 of the ENGOT HRD European initiative the NOGGO GIS v1 assay was evaluated on a further 383 blinded patient samples this time including *BRCA1*- and *BRCA2*-mutated patients [32]. Again, samples were processed as described above and NOGGO GIS status as well as *BRCA* mutation status were reported to the ENGOT for statistical analysis. As in the PAOLA-1 trial patients were considered HRD-positive when either GI-positive and/or harboring a pathological *BRCA1* or *BRCA2* mutation. PFS showed comparable hazard ratios (HR) for Myriad MyChoice (HR 0.352) and the NOGGO GIS v1 assay (HR 0.310) for patients classified as HRD (Figure 4). For HRP patients the HR were (HR 1.128) and (HR 1.023) for Myriad MyChoice and the NOGGO GIS v1 assay, respectively. Similarly, OS data with HR for Myriad MyChoice of 0.5 compared with 0.370 for the NOGGO GIS v1 assay were found.

To compare the assays independently of *BRCA1/2* mutation status, the subgroup of *BRCA1/2* wild-type patients was also analyzed. HR in this subgroup was comparable with 0.364 vs. 0.346 for Myriad MyChoice and the NOGGO GIS v1 assay, respectively.

While Myriad MyChoice was unevaluable in 44 patients the dropout rate for the NOGGO GIS v1 assay was much lower with only 16 samples that could not be evaluated.

To assess whether the 44 cases unevaluable with Myriad MyChoice gave a clinically meaningful result with the NOGGO GIS v1 assay, Kaplan–Meier curves were plotted for this small subgroup and showed a similar pattern, even though statistical analysis was not possible in this small subgroup (Figure 5).

## 4. Discussion

### 4.1. Aim and Assay Content

With the approval of maintenance therapy with olaparib and bevacizumab for high-grade ovarian cancer patients with a deficient homologous recombination pathway, the need for proper genomic testing emerged that not only enables the detection of *BRCA1* and *BRCA2* mutations, but also the evaluation of genomic instability. Since a central laboratory testing approach as favored by Myriad poses great difficulties logistically as well as from the reimbursement point of view, we aimed to design a comprehensive, hybrid-capture NGS-based assay, that can be run locally and addresses currently foreseeable clinical needs.

So far, HRD has been described not only in ovarian and breast cancer, but also in pancreatic and prostate cancer. For this reason, our assay design comprised several modules that served distinct functions. In addition to the essential analysis of *BRCA1/2* mutations, which included the detection of large deletions and complex mutations, the assay also allows for mutation detection of known HRR genes, which have been found to be relevant to prostate cancer [39]. Additionally, the assay encompassed mutational analysis of several oncogenes and tumor suppressor genes, such as *PIK3CA*, *NRAS*, *KRAS*, *AR* and mismatch repair genes. For use in breast cancer patients, the full exonic territory of *HER2* and *ESR1* was included to allow for mutation as well as *HER2* amplification detection.

### 4.2. SNP Backbone and Assay Flexibility (Research Groups)

The calculation of the NOGGO GIS is based solely on the OneSeq CNV Backbone (Agilent Technologies) that can easily be incorporated into any Agilent HS2 hybrid capture custom design. This provides the flexibility to customize the list of enriched genes according to future clinical requirements, without the need to reassess the performance of GIS calculation. Therefore, we believe that the NOGGO GIS v1 assay is unique in comparison to many other available assays, as it addresses a large number of clinically relevant questions with the option to modify the content for future inquiries, making it particularly suitable for research groups such as NOGGO.

### 4.3. Dropout Low Rate

When the bioinformatic pipeline was devised, a conscious decision was made to examine changes of LOH based on the territory that could actually be sequenced, rather than the entire genome. Our expectation was that this would allow us to compensate for poor DNA quality, or areas of the genome that could only be poorly sequenced to increase the robustness of the assay. Interestingly, we could show that in comparison to Myriad MyChoice assay, the dropout rate for the PAOLA-1 clinical samples was significantly lower (*n* = 44 vs. *n* = 16 out of 383, respectively), suggesting that in a routine clinical setting a reduced dropout rate can be expected.

Interestingly, for the subcohort of 44 cases that could only be analyzed with NOGGO GIS v1 assay but not with the Myriad MyChoice assay, the rates of survival suggest that a clear patient stratification is possible. This finding implies that the NOGGO GIS v1 assay could be used to stratify patients based on their clinical outcomes and that it may offer a valuable alternative to the Myriad MyChoice assay (Figure 5). Equally, we observed a relatively low dropout rate in clinical practice since the implementation of the NOGGO GIS v1 assay.

This being an academic initiative, it was important that the bioinformatic was based on publicly and openly available toolkit. The custom script based on publicly available analysis software is available on request.

Since the development of the NOGGO assay, several commercially available HRD assays appeared on the market where bridging data to Myriad MyChoice has been also shown, including the assays mentioned by the German Harmonization Study [32].

We feel that relying solely on bridging data to the Myriad MyChoice assay is not sufficient to demonstrate the clinical performance and utility of molecular diagnostic assays that cover complex biomarkers such as HRD. Many comparative studies between commercially available assay and Myriad MyChoice have been made, but often clinical assessment is lacking [40]. This is particularly important given the intricate nature of HRD assays and the need for them to be thoroughly evaluated to ensure their efficacy in a clinical setting, also reviewed in [31,41]. The ENGOT HRD initiative that made samples of the ENGOT PAOLA-1 study available for assay validation has proven to be an invaluable resource to the scientific community, as it has facilitated the validation of several HRD assays, thereby paving the way for more precise and personalized treatment approaches.

To the best of our knowledge, our study represents the first academic attempt to analyze genomic instability in high-grade serous ovarian cancer using a commercially available, centrally feasible, customized NGS workflow that includes a comprehensive set of HRR targets (encompassing both mutations and structural alterations), as well as a publicly available bioinformatics pipeline that yields a clinically validated cutoff. With HRD testing being a pivotal requirement for informing future therapy decisions, our assay expands the potential application of PARPi by enabling the identification of patient subgroups that stand to benefit the most from this form of treatment.

The current retrospective data underscore the clinical utility of the NOGGO GIS v1 assay. A prospective validation is planned as part of the NOGGO-ENGOT N Plus clinical trial with the goal of reduction of chemotherapy duration in the era of maintenance therapy with PARP-inhibitors.

## 5. Conclusions

Here we have described the NOGGO GIS v1 assay for HRD detection that was validated clinically on ENGOT PAOLA-1 clinical trial samples with PFS and OS data comparable to the Myriad MyChoice assay used in the PAOLA-1 clinical trial. The NOGGO GIS V1 assay is a modular assay with a low drop out rate that not only determines the genome instability status but also alterations like SNVs, InDels and copy number in further relevant cancer related genes.

## Figures and Tables

**Figure 1 cancers-15-03445-f001:**
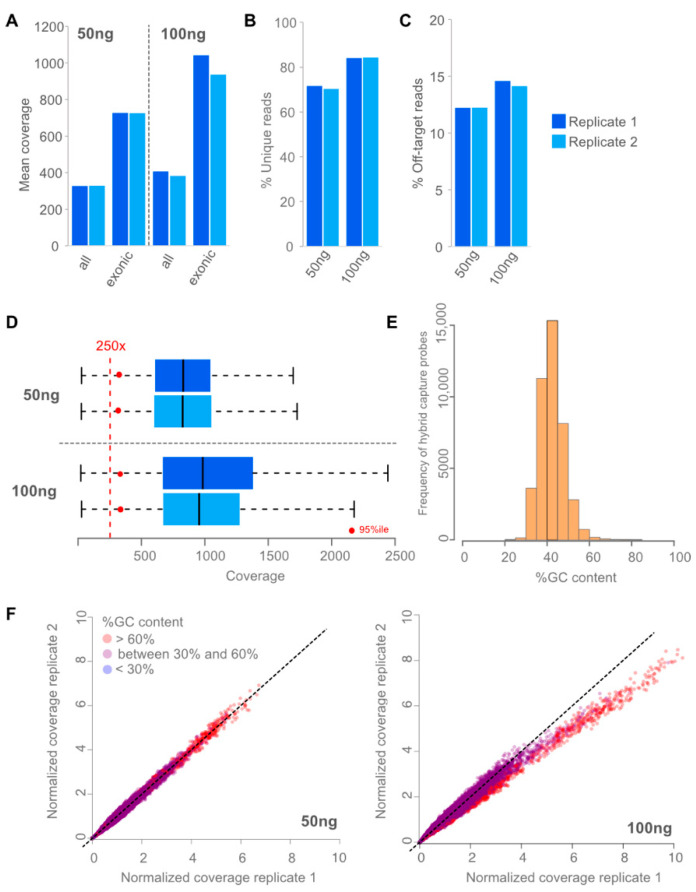
Performance of assay. (**A**): Mean coverages reached using 50 ng and 100 ng of input DNA and sequencing 20 million read pairs in two independent replicates. As intended, mean coverages are twice as high in genes (exonic bases) compared to mean coverage in all targets (all bases). Using 100 ng input led to slightly higher mean coverages. (**B**): Percentage of unique reads for 50 and 100 ng of input DNA are shown in two independent replicates. Using 100 ng input DNA led to slightly higher complexity of the library measured in fraction of unique reads. (**C**): Percentage of Off-target reads are shown for two independent replicates. Off-target rates were stable and low (<20%) leading to efficient usage of the data. Using 100 ng of input led to slightly more off-target reads. (**D**): Coverage distribution of exonic positions in two independent replicates (dark and light blue, respectively). Blueboxes represent the 75% percentile with the median coverage depicted as black line. 50 ng (**top**) and 100 ng (**bottom**) DNA input are shown. Red dots represent the 95 percentile; hence, 95% of exonic positions had a coverage greater than the red dot. (**E**): Histogram of capture probes GC content in 5%tile bins are shown. (**F**): Normalized deduplicated coverage correlation of experimental replicates with 50 ng (**left**) and 100 ng (**right**) are shown. Colors from blue over purple to red reflect the degree of GC content (blue: low, red: high).

**Figure 2 cancers-15-03445-f002:**
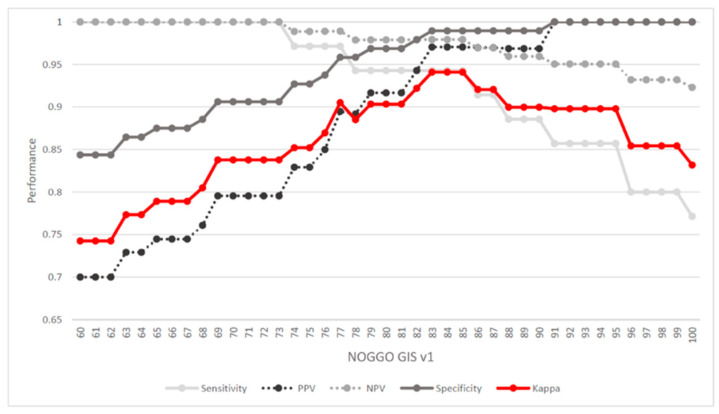
Cutoff determination: Cohen’s kappa coefficient was used to determine the maximum correspondence between Myriad MyChoice HRD status and NOGGO GIS (red line). In addition, sensitivity PPV, specificity, and NPV were calculated. These measures indicated that a NOGGO GIS cutoff between 83 and 85 led to the maximum correspondence between the two scores. PPV: positive predicted value; NPV: negative predicted value.

**Figure 3 cancers-15-03445-f003:**
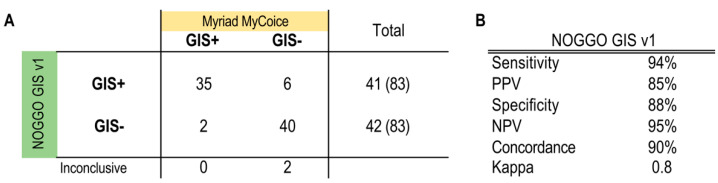
Results of the phase 2 blinded testing of 85 tumor samples from PAOLA-1 *BRCA* wild-type patients. (**A**) The numbers of true positives, false positives, false negatives and true negatives according to the Myriad MyChoice test are listed. (**B**) Calculated performance statistics using the numbers from the table A GIS: Genomic instability score; PPV: positive predicted value; NPV: negative predicted value.

**Figure 4 cancers-15-03445-f004:**
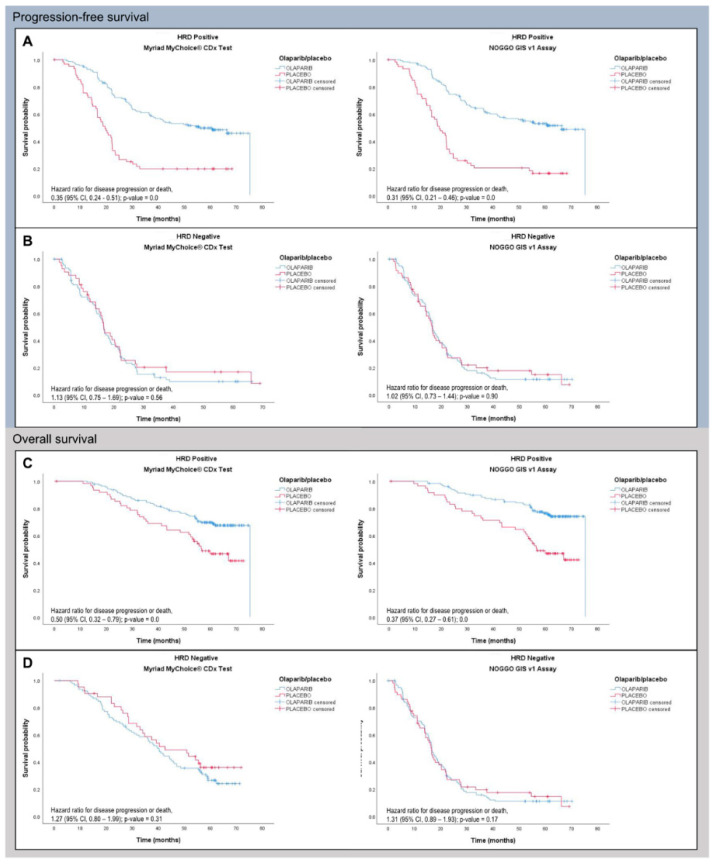
PFS (**A**,**B**) and OS (**C**,**D**) data of 383 patients of the PAOLA-1 clinical trial analyzed with either Myriad MyChoice (cutoff ≥ 42) or the NOGGO GIS v1 assay (cutoff ≥ 83).

**Figure 5 cancers-15-03445-f005:**
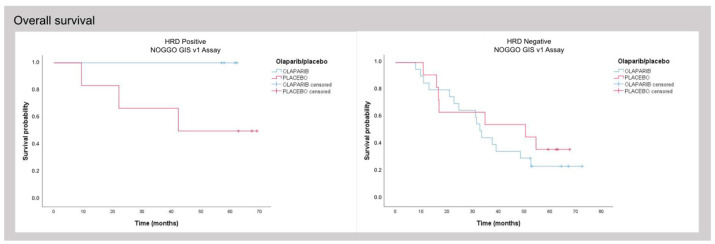
Subgroup analysis of patients successfully analyzed with the NOGGO GIS v1 assay for which no results with Myriad MyChoice could be obtained.

**Table 1 cancers-15-03445-t001:** Genes list of the NOGGO GIS v1 Assay. List of genes examined for point mutations, insertions/deletions (InDels) and copy number changes including relevant HRR genes and driver mutations. * HRR genes; # genes for detection of copy number changes; bold genes for driver mutation detection.

*ABRAXAS1*	*BRIP1 **	*FANCC **	*MRE11A **	*RAD51B **
*APC*	*BUB1B*	*FANCD2 **	*MSH2*	*RAD51C **
*AR*	*CDH1*	*FANCE **	*MSH6*	*RAD51D **
*ARID1A **	*CDK12 **	*FANCF **	*NBN **	*RAD52 **
*ATM **	*CHEK1 **	*FANCG*	** *NRAS* **	*RAD54L **
*ATR **	*CHEK2 **	*FANCI **	*PALB2 **	*RPA1 **
*ATRX*	*CTNNB1*	*FANCL **	** *PIK3CA* **	*STK11*
*BARD1 **	** *EGFR ^#^* **	*FANCM **	*PMS2*	*TP53*
*BLM **	*EMSY *^#^*	*HDAC2 **	*PPP2A2R*	*XRCC2 **
** *BRAF* **	** *ERBB2 ^#^* **	*HOXB3*	*PTEN ^#^*	
*BRCA1 *^#^*	*ESR1*	** *KRAS* **	*RAD50 **	
*BRCA2 *^#^*	*FANCA **	*MLH1 **	*RAD51 **	

## Data Availability

All required tools are publicly available and can be downloaded with relevant manuals. The custom PERL script is available under GNU license and a link can be requested by the corresponding authors.

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
