# Peer review of "Development of the NOGGO GIS v1 Assay, a Comprehensive Hybrid-Capture-Based NGS Assay for Therapeutic Stratification of Homologous Repair Deficiency Driven Tumors and Clinical Validation"

_cancers, 2023, doi:10.3390/cancers15133445_

Round 1

Reviewer 1 Report

The paper deals with a very relevant clinical topic, identification of HRD tumors for therapeutic stratification.The study is timing and clinically relevant.

Material and Methods.

Lines 142-143. “was designed targeting all exonic bases of selected genes” 

Is this including all 5’ and 3’ UTR bases? Exon-intron boundaries not included? If not, what about splice site variants? Clarify

Line 169. “information on LOH was used as a pathogenicity factor for both oncogenes and tumor suppressor genes”

Please, clarify the use of LOH as a pathogenic factor for oncogenes.

Line 195-196 “PLOH, when larger than 10 Mb”

What if LOH >10Mb includes a whole chromosome or chromosome arm? Counts as PLOH?

I recommend adding to the methods section the following:

Refer to the DNA integrity analysis performed in the study. Method? Average integrity score in the 85+383 samples?

How the 85 PAOLA-1 BRCA negative samples were selected? (Not fully random, since all them BRCA negative). Please, clarify

How the phase 3 383 PAOLA-1 samples were selected? (fully random?)

Results

Lines 293-295 “To compare the assays independently of BRCA1/ 2 mutation status the subgroup of 293 BRCA1/2 wild type patients was also analyzed. HR in this subgroup were comparable 294 with 0.364 vs. 0.346 for Myriad MyChoice and NOGGO GIS v1 assay, respectively”

I think the above is highly relevant. I will appreciate a specific figure to show this analysis (additional figure, supplemental figure or modification of current figure 3).  

Lines 299-300 “While Myriad MyChoice was unevaluable in 44 patients the dropout rate for the 299 NOGGO GIS v1 assay was much lower with only 16 samples that could not be evaluated”

I recommend adding to the result section the following:

Any overlap between the 44 and 16 dropout samples. Clarify. If there is an overlap, exactly same block in all cases? (Specify)

In relation with the phase-3 383 PAOLA-1 samples, I will appreciate a comparative analysis of Myriad MyChoice and NOGGO GIS v1 in relation with BRCA status (perhaps as supplementary data)

Same variants identified by both methods?

Same variant classification?

Discussion

Line 320-321

Please, specify at least one example of BRCA1/2 large deletion an BRCA1/2 “complex” mutation detected by NOGGO GIS v1 in FFPE samples.

Line 322

Please, specify some HRR genes specifically relevant to prostate cancer.

Line 325

Explain the meaning of “full exonic territory”

Line 327

The authors suggest as a possible explanation for the low dropout rate “a conscious decision was made to examine changes of LOH based on the territory that could actually be sequenced, rather than the entire genome”.

The explanation makes sense to me but, intuitively, I would think that the smaller the territory, the less accurate the score.  Can the author’s comment on this?

In addition, can the authors comment on the following topic. In their opinion, the excess of dropouts observed with Myriad MyChoice is observed only in poor quality samples (low DNA integrity)?  In samples with good DNA integrity but low percentage of tumor content? in both?

Minor points.

Line 59-60.    The author mention “full exonic territory of relevant cancer oncogenes 59 and HRR genes” I think some of the target genes (e.g. APC) are neither oncogenes nor HRR genes. Modify as needed.

Author Response

Hamburg, 08.06.2023

Dear reviewing experts,

thank you for considering our manuscript “Development of a Comprehensive Hybrid Capture Based NGS Assay for Therapeutic Stratification of Homologous Repair Deficiency Driven Tumors and Clinical Validation” for publication in Cancers. We thank the reviewers for their detailed and constructive comments and for the helpful suggestions. We have addressed each reviewer´s concerns point by point in this letter and have corrected the manuscript accordingly using “Track Changes” function.

Thank you and best regard,

Lukas Heukamp

Dear Reviewer 1,

thank you for your in depths evaluation and your very valid points of concern. We would like to address them one by one:

  • Lines 142-143. “was designed targeting all exonic bases of selected genes” Is this including all 5’ and 3’ UTR bases? Exon-intron boundaries not included? If not, what about splice site variants? Clarify

The assay design covers at least 10bp of adjacent intronic region, in most cases due to the Hybrid capture design at least another 10 bp are covered. Since DNA fragments include intronic regions in most cases much more intronic territory will be sequenced. 

We have changed the wording to: including the adjacent intronic regions of at least 10 bp.

  • Line 169. “information on LOH was used as a pathogenicity factor for both oncogenes and tumor suppressor genes” Please, clarify the use of LOH as a pathogenic factor for oncogenes.

Thank you for this comment. We have clarified the text accordingly. According the classification guidelines by Koeppel et al 2021 (PMID: 34700215, Table 8 or Chapter 3.4.3.) the co-occurrence of a certain mutation in an oncogene with an amplification/LOH can be used as an additional, moderate evidence of pathogenicity (PM7). This information can be useful for rare, hotspot-region mutations in oncogenes, such as PIK3CA or ERBB2.

We can assess LOH for all genes covered in the assay and this is particularity relevant for the tumor suppressor genes including the HRR genes. We sometimes also see LOH for oncogenes such as KRAS and EGFR as well as amplifications of the mutated allele. This is relevant for example in lung cancer where high allele frequencies of mutates EGFR alleles are frequently observed. As this is not as relevant in ovarian cancer we have simplified the text.

  • Line 195-196 “PLOH, when larger than 10 Mb” What if LOH >10Mb includes a whole chromosome or chromosome arm? Counts as PLOH?

Thank you for this very relevant point. We have added “and smaller than a complete chromosome arm“ to the text.

  • I recommend adding to the methods section the following: Refer to the DNA integrity analysis performed in the study. Method? Average integrity score in the 85+383 samples?

The DNA fo the PAOLA-1 samples was not checked for DNA quality by the ENGOT central lab before sending samples to the participating labs of the initiative.

As a receiving lab for the PAOLA-1 samples we only got very limited material so that we did not perform a Tape station analysis for theses samples. We have added the requested information to the text. In routine application of the NOGGO- GIS V1 Assay we asses DNA integrity using the Agilent Tape station. However, usually samples are not excluded from analysis as there is a clinical need for obtaining a result for a particular patient and alternative material of better quality is usually not available.

How the 85 PAOLA-1 BRCA negative samples were selected? (Not fully random, since all them BRCA negative). Please, clarify

Samples were selected on wild type BRCA status as determined by Myriad MyChoice as part of the PAOLA1 trial and based on tissue availability.

  • Lines 293-295 “To compare the assays independently of BRCA1/ 2 mutation status the subgroup of 293 BRCA1/2 wild type patients was also analyzed. HR in this subgroup were comparable 294 with 0.364 vs. 0.346 for Myriad MyChoice and NOGGO GIS v1 assay, respectively”. I think the above is highly relevant. I will appreciate a specific figure to show this analysis (additional figure, supplemental figure or modification of current figure 3).

Thank you for your interest in this data. We will include this following figure in the supplemental data.

  • Lines 299-300 “While Myriad MyChoice was unevaluable in 44 patients the dropout rate for the 299 NOGGO GIS v1 assay was much lower with only 16 samples that could not be evaluated”. I recommend adding to the result section the following: Any overlap between the 44 and 16 dropout samples. Clarify. If there is an overlap, exactly same block in all cases? (Specify)

Interestingly, there was no overlap between the 44 and 16 dropout samples.

  • In relation with the phase-3 383 PAOLA-1 samples, I will appreciate a comparative analysis of Myriad MyChoice and NOGGO GIS v1 in relation with BRCA status (perhaps as supplementary data). Same variants identified by both methods? Same variant classification?

We agree that this is highly relevant. Sadly, only BRCA mutation status given as + or – was available for the Myriad data and thus a direct comparison of BRCA mutation calling by Myriad and the NOGGO Assay is not possible. We will try to address BRCA1/2 mutation calling strategies in conjunction with the NOGGO GIS v1 assay  in a follow-up white paper.

  • Line 320-321; Please, specify at least one example of BRCA1/2 large deletion an BRCA1/2 “complex” mutation detected by NOGGO GIS v1 in FFPE samples.

Thank you for this question. We feel that the detection of complex BRCA1/2 alterations is an imported user requirement for the NOGGO GIS v1 assay. To be able to do this both the capture of relevant regions by the hybrid capture design  (part of the NOGGO GIS v1 assay) as well as appropriate variant calling on the bioinformatic sides are needed.  We use JuLI for fusion and PureCN for CNV calling that will also detect Exon deletions.

Although we have found several of these alterations in our routine work, none was observed as part of the PAOLA-1 samples. We will describe some of these cases in a future separate publication.

  • Line 322; Please, specify some HRR genes specifically relevant to prostate cancer.

For example, ATM. This topic is reviewed in for example in Scott et al (doi: 10.18632/oncotarget.28015). We have included this reference in the publication.

  • Line 325; Explain the meaning of “full exonic territory”

Covering all coding bases of all known transcripts accord to NCBI as well as a minimum of 10bp flanking region.

  • Line 327; The authors suggest as a possible explanation for the low dropout rate “a conscious decision was made to examine changes of LOH based on the territory that could actually be sequenced, rather than the entire genome”. The explanation makes sense to me but, intuitively, I would think that the smaller the territory, the less accurate the score. Can the author’s comment on this?

This is a highly relevant point. We have used in silico modelling to determine how reduction in the number of SNPs affects the HRD score and found even a loss of more than 50% still gave reliable results.

It is also possible the some cases were not analyzable by Myriad due to low tumor cell content. Since we were supplied with DNA only and did not assess the tumor content on FFPE slides, we can only judge the tumor content from bioinformatic purity estimation.

The mean purity of the 44 Drop out cases was somewhat lower compared to the rest of the cohort.

  • Line 59-60; The author mention “full exonic territory of relevant cancer oncogenes 59 and HRR genes” I think some of the target genes (e.g. APC) are neither oncogenes nor HRR genes. Modify as needed.

We have changed “oncogenes” to “cancer related genes” in the text to include APC, a non HRR tumor suppressor gene.

Reviewer 2 Report

 The most common and deadly type of ovarian cancer is the epithelial-derived high grade serous ovarian carcinoma (HGSOC). In 2018, the approach to the treatment of patients with advanced OC changed when the results of the SOLO1 trial allowed registration of olaparib in maintenance treatment after the first line in patients with stages III-IV with pathogenic BRCA1/2 mutation in the case of the effect of platinum-containing therapy.

The most recent NCCN guidelines recommend somatic testing in the up-front setting for BRCA1/2 mutations, NTRK fusions, homologous recombination deficiency (HRD) and  tumor biomarkers including microsatellite instability (MSI), mismatch repair deficiency (MMR) and tumor mutation burden (TMB) for all patients [National Comprehensive Cancer Network. Available online: https://www.nccn.org/professionals/physician_gls/pdf/ovarian.pdf (accessed on 8 September 2021).].

Currently, the only two commercial tests approved by the United States Food and Drug Administration (FDA) for assessing the status of HRD based on SNPs are the myChoice CDx (Myriad Genetics) and FoundationFocus™ CDxBRCA LOH (Foundation Medicine) assays.

There is an urgent need  to develop reliable, accessible, standardized   assay with minimized costs to detect violations in the system of homologous repair, genomic instability , in order to predict sensitivity to PARPi and find optimal therapy for OC patients.

For me (who is not a specialist in NGS sequencing)  the methods are described in details and the results are clearly presented. 

But I would like to ask authors to add to the discussion the recent papers, where the other tests for HRD status of OC patients are described.

-Wenbin Li, Lin Gao, Xin Yi, Shuangfeng Shi, Jie Huang, Leming Shi, Xiaoyan Zhou, Lingying Wu, Jianming Ying, Patient Assessment and Therapy Planning Based on Homologous Recombination Repair Deficiency. Genomics, Proteomics & Bioinformatics. 2023 . doi.org/10.1016/j.gpb.2023.02.004.

(https://www.sciencedirect.com/science/article/pii/S1672022923000360)

-Mangogna A, Munari G, Pepe F, Maffii E, Giampaolino P, Ricci G, Fassan M, Malapelle U, Biffi S. Homologous Recombination Deficiency in Ovarian Cancer: from the Biological Rationale to Current Diagnostic Approaches. J Pers Med. 2023 Feb 2;13(2):284. doi: 10.3390/jpm13020284. PMID: 36836518; PMCID: PMC9968181.

-Magliacane G, Brunetto E, Calzavara S, Bergamini A, Pipitone GB, Marra G, Redegalli M, Grassini G, Rabaiotti E, Taccagni G, Pecciarini L, Carrera P, Mangili G, Doglioni C, Cangi MG. Locally Performed HRD Testing for Ovarian Cancer? Yes, We Can! Cancers (Basel). 2022 Dec 21;15(1):43. doi: 10.3390/cancers15010043. PMID: 36612041; PMCID: PMC9817883.

The paper can be published after minor revision

line 108 through instead of though

line 352  being  instead of this being

Author Response

Hamburg, 08.06.2023

Dear reviewing experts,

thank you for considering our manuscript “Development of a Comprehensive Hybrid Capture Based NGS Assay for Therapeutic Stratification of Homologous Repair Deficiency Driven Tumors and Clinical Validation” for publication in Cancers. We thank the reviewers for their detailed and constructive comments and for the helpful suggestions. We have addressed each reviewer´s concerns point by point in this letter and have corrected the manuscript accordingly using “Track Changes” function.

Thank you and best regards,

Lukas Heukamp

Dear Reviewer 2,

thank you very much for your kind remarks and for your constructive comments.

We have read the suggested publications and feel that they offer much additional insight and summary to the topic. We have included the refences in the manuscript.

Reviewer 3 Report

The manuscript describes a sequencing-based test (NOGGO GIS v1 test) able to distinguish between HRD-positive versus HRD-negative patients suffering from high grade serous ovarian cancer treated with olaparib, a PARP inhibitor. The results demonstrates the accuracy of the test in retrospective analysis using DNA from around 400 FFPE samples included in the PAOLA-1 clinical trial. The test was compared with Myriad MyChoice, and displayed lower dropout rate. Potentially, the test has clinical utility as it showed robust hazard ratios for progression-free survival (PFS) and overall survival (OS). The test stratifies patients depending on the GI-score, which was based on i) percent loss of heterozygosity (PLOH), ii) percent copy number alteration (PCNA), and iii) percent telomeric copy number alterations (PCTCNA). Before publication, the manuscript needs some minor changes.

-          Description of the panel is adequate. Inclusion of probes to detect mutations in HRR genes and driver mutations should provide important information about the pathogenicity of the detected variants. However, no information about the variants found in the clinical samples analyzed are shown, neither their possible correlation with the patient outcome during the PAOLA-1 clinical trial. The impact of the manuscript will be much higher if the authors could provide information about the most prevalent variants in HRR genes and driver mutations detected, and their association with GI-score, and hazard ratios for PFS and OS.

-          In relation with the methods, a deeper description of the custom workflow used for mutation calling would be necessary: free software tools names, versions and references.

-          Authors claim at the end of the discussion (lines 369-370) that the study represents “a publicly available bioinformatics pipeline that yields a clinically validated cutoff”. Although the authors described with many details the pipeline, it would be much better if the full details of the pipeline is uploaded to a open source software repository such as GitHub or similar.

-          Figure 1 description is very poor. Panel A: Please show the Standard Error of the Mean (SEM) coverages or the Standard Deviation to understand the variability of then Mean Coverages. Panel B and C: Please, explain properly what are the Y axis Titles. Panel D: the image is cropped; please explain what is the Y axis Titles, and what are the differences between light versus dark grey boxes. Panel E and F: please explain properly what is represented in the X and Y axes. Panel G: the image is cropped, and cannot be understood.

English is OK,

Author Response

Hamburg, 08.06.2023

Dear reviewing experts,

thank you for considering our manuscript “Development of a Comprehensive Hybrid Capture Based NGS Assay for Therapeutic Stratification of Homologous Repair Deficiency Driven Tumors and Clinical Validation” for publication in Cancers. We thank the reviewers for their detailed and constructive comments and for the helpful suggestions. We have addressed each reviewer´s concerns point by point in this letter and have corrected the manuscript accordingly using “Track Changes” function.

Thank you and best regards,

Lukas Heukamp

Dear Reviewer 3,

thank you for your detailed revision, we would like to address your comments one by one:

  • Description of the panel is adequate. Inclusion of probes to detect mutations in HRR genes and driver mutations should provide important information about the pathogenicity of the detected variants. However, no information about the variants found in the clinical samples analyzed are shown, neither their possible correlation with the patient outcome during the PAOLA-1 clinical trial. The impact of the manuscript will be much higher if the authors could provide information about the most prevalent variants in HRR genes and driver mutations detected, and their association with GI-score, and hazard ratios for PFS and OS.

Thank you for this comment. We also feel that HRD assessment is very useful in determining the pathogenicity of HRR and BRCA1 and BRCA2 variants. The scope of this publication is the description of the HRD algorithm and clinical validation of the NOGGO GIS v1 score. We plan to look at the HRR Variants of the PAOLA-1 cohort as well as our routine clinical samples in a more comprehensive follow-up publication. Furthermore, we plan to set-up a registry of these variants in the context of HRD as part of that future publication.

  • In relation with the methods, a deeper description of the custom workflow used for mutation calling would be necessary: free software tools names, versions and references.

We have included the details of the tools used in the text lines 177-183 and that  are publicly available. Related publications are to be found on the respective download webpages and documentation. i.e for PureCN. https://github.com/lima1/PureCN

Our custom script for implementing the NOGGO GIS v1 score is available also via GIT Hub to interested parties on request under a GNU license. We have given contact details in the text.

  • Authors claim at the end of the discussion (lines 369-370) that the study represents “a publicly available bioinformatics pipeline that yields a clinically validated cutoff”. Although the authors described with many details the pipeline, it would be much better if the full details of the pipeline is uploaded to a open source software repository such as GitHub or similar.

We will make the script available on GitHub and access will be granted with GNU License upon request to the corresponding author. See line 392

Furthermore, the NOGGO GIS v1 pipeline will be available as part of the Agilent Alissa bioinformatics suite, probably by the end of this year. Since we are not able to publicly speak for Agilent, we did not included this information in the publication.

  • Figure 1 description is very poor. Panel A: Please show the Standard Error of the Mean (SEM) coverages or the Standard Deviation to understand the variability of then Mean Coverages. Panel B and C: Please, explain properly what are the Y axis Titles. Panel D: the image is cropped; please explain what is the Y axis Titles, and what are the differences between light versus dark grey boxes. Panel E and F: please explain properly what is represented in the X and Y axes. Panel G: the image is cropped, and cannot be understood.

Thank you for this absolutely valid point. We have rewritten the figure legend and improved the labels of the figure.